# Estimation of benzathine penicillin G demand for congenital syphilis elimination with adoption of dual HIV/syphilis rapid diagnostic tests in eleven high burden countries

**Sapan Shah**[1]*, **Surbhi Garg**[1], **Katherine Heath**[2,3], **Obiageli Ofili**[4], **Yashika Bansal**[1], **Frederic Seghers**[4], **Andrew Storey**[4], **Melanie Taylor**[2,5]

**1** Clinton Health Access Initiative (CHAI) India, New Delhi, Delhi, India, **2** Department of Global Programmes of HIV, Hepatitis, STI, World Health Organization (WHO), Geneva, Switzerland, **3** Macfarlane Burnet Institute, Melbourne, Australia, **4** Clinton Health Access Initiative (CHAI), Boston, Massachusetts, United States of America, **5** United States Centers for Disease Control and Prevention, Atlanta, GA, United States of America

* shah.sapan11@gmail.com

**Data Availability Statement:** All relevant data are within the manuscript and its Supporting Information files.

## Abstract

### Background

WHO recommends use of rapid dual HIV/syphilis tests for screening pregnant women (PW) during antenatal care to prevent mother-to-child transmission. Scale-up of testing implies a need to accurately forecast and procure benzathine penicillin (BPG) to treat the additionally identified PW with syphilis.

### Methods

Country-reported ANC coverage, PW syphilis screening and treatment coverage values in 2019 were scaled linearly to EMTCT targets by 2030 (constant increasing slope from 2019 figures to 95% in 2030) for 11 focus countries. Antenatal syphilis screening coverage was substituted with HIV screening coverage to estimate potential contribution of rapid dual HIV/syphilis tests in identifying additional PW with syphilis. BPG demand was calculated for 2019–2030 accordingly.

### Results

The estimated demand for BPG (in 2.4 million unit vials) using current maternal syphilis prevalence and treatment coverage will increase from a baseline of 414,459 doses in 2019 to 683,067 doses (+65%) in 2021 assuming immediate replacement of single HIV test kits with rapid dual HIV/syphilis tests for these 11 countries. Continued scale up of syphilis screening and treatment coverage to reach elimination coverage of 95% will result in an estimated demand increase of 160%, (663,969 doses) from 2019 baseline for a total demand of 1,078,428 BPG doses by 2030.

**Funding:** Sapan Shah (SS), Surbhi Garg (SG) and Yashika Bansal (YB) are on the payroll funded by the Bill & Melinda Gates Foundation. The grant number is OPP1203172, and the website is https://www.gatesfoundation.org/ The funders had no role in study design, data collection and analysis, decision to publish, or preparation of the manuscript.

**Competing interests:** The authors have declared that no competing interests exist.

## Conclusions

Demand for BPG will increase following adoption of rapid dual HIV/syphilis test kits due to increases in maternal diagnoses of syphilis. To eliminate congenital syphilis, MNCH clinical programs will need to synergize with disease surveillance programs to accurately forecast BPG demand with scale up of antenatal syphilis screening to ensure adequate treatment is available for pregnant women diagnosed with syphilis.

## Background

Approximately one million pregnant women (PW) had active syphilis in 2016, resulting in over 600,000 cases of congenital syphilis and greater than 350,000 adverse birth outcomes according to World Health Organization [1]. Maternal syphilis can result in adverse birth outcomes in >50% of untreated pregnancies, including stillbirth, congenital syphilis, prematurity or low-birth weight, congenital deformities, and neonatal death [2]. Injectable benzathine benzylpenicillin, or benzathine penicillin G (BPG) is the only recommended treatment that, if provided early in pregnancy, can prevent mother-to-child transmission (MTCT) of syphilis and resulting adverse birth outcomes [3]. Other drugs are contraindicated, do not cross the placenta to treat the foetus or do not have published evidence of clinical efficacy in prevention of congenital syphilis [4]. Additionally, there is no documented penicillin resistance in *Treponema pallidum* (the causative bacterial agent for syphilis) [5].

The WHO recommends that all PW be tested for syphilis early in pregnancy as part of antenatal care (ANC) [4]. WHO has established targets for country validation of elimination of mother-to-child transmission (EMTCT) of syphilis which include: (a) at least 95% of PW attend ANC at least once, (b) at least 95% of PW receive syphilis screening during ANC and (c) at least 95% of syphilis seropositive PW receive adequate treatment, defined as at least one intramuscular injection of 2.4 million units (MU) of BPG [6]. Globally, over one-third of PW attending ANC (37%) were not tested for syphilis in 2016 [7], despite diagnosis and subsequent prevention of MTCT of syphilis being feasible, inexpensive, and cost-effective [8]. Use of rapid syphilis tests has resulted in substantial increases in antenatal syphilis screening in all settings, even in facilities with low rates of coverage [9].

In 2019, WHO recommended the rapid dual HIV/syphilis test as the first-line test for PW attending antenatal care (ANC) [10]. Dual HIV/syphilis RDTs have shown high levels of sensitivity and specificity for HIV and syphilis in ANC settings [11]. Integrating maternal syphilis screening and treatment into HIV EMTCT programmes using RDTs is a simple and cost-effective intervention for global health investment and minimises adverse pregnancy outcomes compared with other testing strategies [12]. In many countries, ~95% of women are tested for HIV during ANC; however, <50% are tested for syphilis. Simultaneous testing for HIV and syphilis could rapidly close the gap between HIV and syphilis screening [13].

The increased level of syphilis screening anticipated for PW with the scale-up of dual HIV/syphilis RDT in low-and middle-income countries will identify maternal syphilis cases that would otherwise have been undetected. These cases will require BPG for treatment of syphilis and prevention of MTCT. Due to recent issues with widespread global shortages of BPG and an unreliable supply, effective forecasting and procurement is imperative to ensure BPG is available for maternal syphilis treatment [14].

The accurate estimation of BPG demand prior to and after dual HIV/Syphilis RDT implementation is crucial to anticipate timely and adequate procurement of BPG alongside procurement of RDT kits. In this study, we estimate the current demand for BPG for EMTCT of syphilis in 11 focus countries, representing 37% of global pregnancies and 38% of all births with congenital syphilis in 2016. It was assumed that the implementation of dual HIV/syphilis RDTs in ANC, reinforced by full adoption of appropriate testing for women who already know their HIV status, would lead to equivalent HIV and syphilis screening rates. Further, we project future increases in BPG demand due to improved screening coverage due to dual HIV/Syphilis RDTs.

## Methods

We compiled estimates of ANC coverage, and HIV and syphilis screening and treatment coverage in ANC between 2012 and 2019 using previously published data [7] and estimates from the Global Health Observatory [15] and for HIV from UNAIDS [16] for 11 focus countries: Democratic Republic of Congo, Ethiopia, India, Indonesia, Kenya, Malawi, Mozambique, Nigeria, Tanzania, Uganda and Zambia (S1 File).

We then extrapolated the service coverage estimates of antenatal screening and treatment coverage for syphilis for the next 10 years with the assumption of achievement of WHO EMTCT targets–95% syphilis screening and 95% treatment coverage in ANC–by 2030. It was assumed that the implementation of dual HIV/syphilis RDTs in ANC would lead to equivalent HIV and syphilis screening rates. Finally, the number of PW who would be syphilis-positive and hence need treatment between 2021 and 2030 was calculated using the available epidemiological data and the methods from WHO's Congenital Syphilis Estimation Tool [17].

## Estimating demand for BPG

The demand for BPG to treat maternal syphilis in 2019 (baseline) 2021, 2025 and 2030 was estimated using methods as below and further described in S1 File.

$$No. of\ 2.4\ MU\ BPG\ vials\ required$$
$$= Estimated\ pregnancies \times ANC\ coverage\ (\%)$$
$$\times Syphilis\ screening\ coverage\ (\%)$$
$$\times Prevalence\ of\ positive\ syphilis\ tests\ in\ PW\ (\%)$$
$$\times Syphilis\ treatment\ coverage\ (\%)$$
$$\times No. of\ doses\ per\ PW\ (3\ in\ this\ study)$$

Number of live births, reported by the United Nations Population Division, is used as an proxy for number of pregnancies in the WHO's Congenital Syphilis Elimination Tool [17]. ANC coverage, syphilis and HIV screening coverage and syphilis treatment coverage are included from the WHO Global Health Observatory which is populated with country-reported data. These values are evaluated as per the linear growth model described in the Methods. Maternal syphilis positivity rate is assumed to be constant at the levels recorded in 2019. Number of doses per PW is also kept constant at three doses per PW, owing to limited access to data indicating number of doses administered in the field. The values are then multiplied according to the formula above to evaluate the demand for BPG in any given year.

## Estimating demand for treatment regimens for infants

The required treatment for infants was estimated as below.

$$No.of\ infants\ requiring\ treatment$$
$$= No.of\ livebirths\ to\ pregnant\ women\ found\ syphilis\ reactive$$
$$- No.of\ pregnant\ women\ provided\ with\ treatment\ using\ BPG$$

The number of livebirths to syphilis positive PW is evaluated by multiplying syphilis prevalence, ANC1 coverage and syphilis screening coverage to total livebirths. The number of PW treated is calculated by multiplying this number with the syphilis treatment coverage.

## Baseline

A baseline was constructed for the 11 focus countries, with the ANC, screening and treatment coverage extrapolated or obtained from the various data sources described (Table 1).

## Scenarios considered

Two scenarios were assumed for this analysis:

a. *scaleup to EMTCT targets without dual testing*- ANC coverage, syphilis screening coverage and syphilis treatment scaled up linearly from 2019 to the target coverage 95% in the target year 2030 regardless of HIV screening coverage, and

b. *dual HIV/syphilis RDT implementation in ANC*- syphilis screening coverage increased to the level of HIV screening coverage for all countries immediately, and then increased linearly to reach EMTCT targets by 2030

In scenario (b), it has been assumed that the dual test would replace all single HIV and syphilis screening tests in 2021. However, the actual transition to dual testing in country

**Table 1. Baseline data.**

| Country | Baseline Values (2019) | | | | | |
|---|---|---|---|---|---|---|
| | Live Births[a] | Syphilis Prevalence[b] | ANC Coverage[b] | HIV Screening Coverage[c] | Syphilis Screening Coverage[d] | Syphilis Treatment Coverage[d] |
| **Democratic Republic of Congo** | 3,485,112 | 1.69% | 90% | 35% | 13% | 100% |
| **Ethiopia** | 3,278,896 | 1.40% | 87% | 90% | 61% | 100% |
| **India** | 25,656,209 | 0.10% | 100% | 79% | 34% | 70% |
| **Indonesia** | 4,858,603 | 1.20% | 95% | 49% | 8% | 59% |
| **Kenya** | 1,631,470 | 0.30% | 95% | 89% | 81% | 100% |
| **Malawi** | 719,240 | 1.60% | 100% | 95% | 82% | 100% |
| **Mozambique** | 1,161,076 | 2.90% | 91% | 95% | 83% | 100% |
| **Nigeria** | 7,516,000 | 0.40% | 66% | 37% | 16% | 81% |
| **Tanzania** | 2,225,339 | 1.70% | 98% | 95% | 73% | 73% |
| **Uganda** | 1,812,193 | 1.90% | 98% | 95% | 87% | 93% |
| **Zambia** | 702,971 | 2.53% | 100% | 89% | 65% | 87% |

[a] From WHO Congenital Syphilis elimination tool [17]

[b] From Korenromp et al [7]

[c] From UNAIDS data [16]

[d] From GHO data [15]

programs could be gradual, and 100% replacement may take several years, particularly if governments do not receive catalytic partner support. The analysis, therefore, represents an upper bound of the potential syphilis screening coverage that can be achieved with the dual testing.

To account for this field reality, a snapshot scenario of the change in BPG demand in the scenario of gradual dual test rollout has been built for one country, India. For this analysis, it was assumed that substitution of HIV tests with dual tests will take place gradually and linearly, with the first phase beginning in 2021 and continuing up to 2025, when all HIV tests in ANC will be replaced with dual tests.

## Results

### BPG demand for treating pregnant women with syphilis

The annual BPG demand for 11 focus countries was estimated from 2019 to 2030 (Fig 1). A snapshot of the demand in the years 2019 (baseline), 2021, 2025 and 2030 (Table 2) indicates that without dual testing, demand for BPG (2.4 MU vials) will increase gradually from a baseline of 414,459 in 2019 to 510,948 in 2021 (+23%). With dual testing, this demand would increase to 683,067 vials in 2021 (+65% over 2019 baseline) when ANC screening coverage for

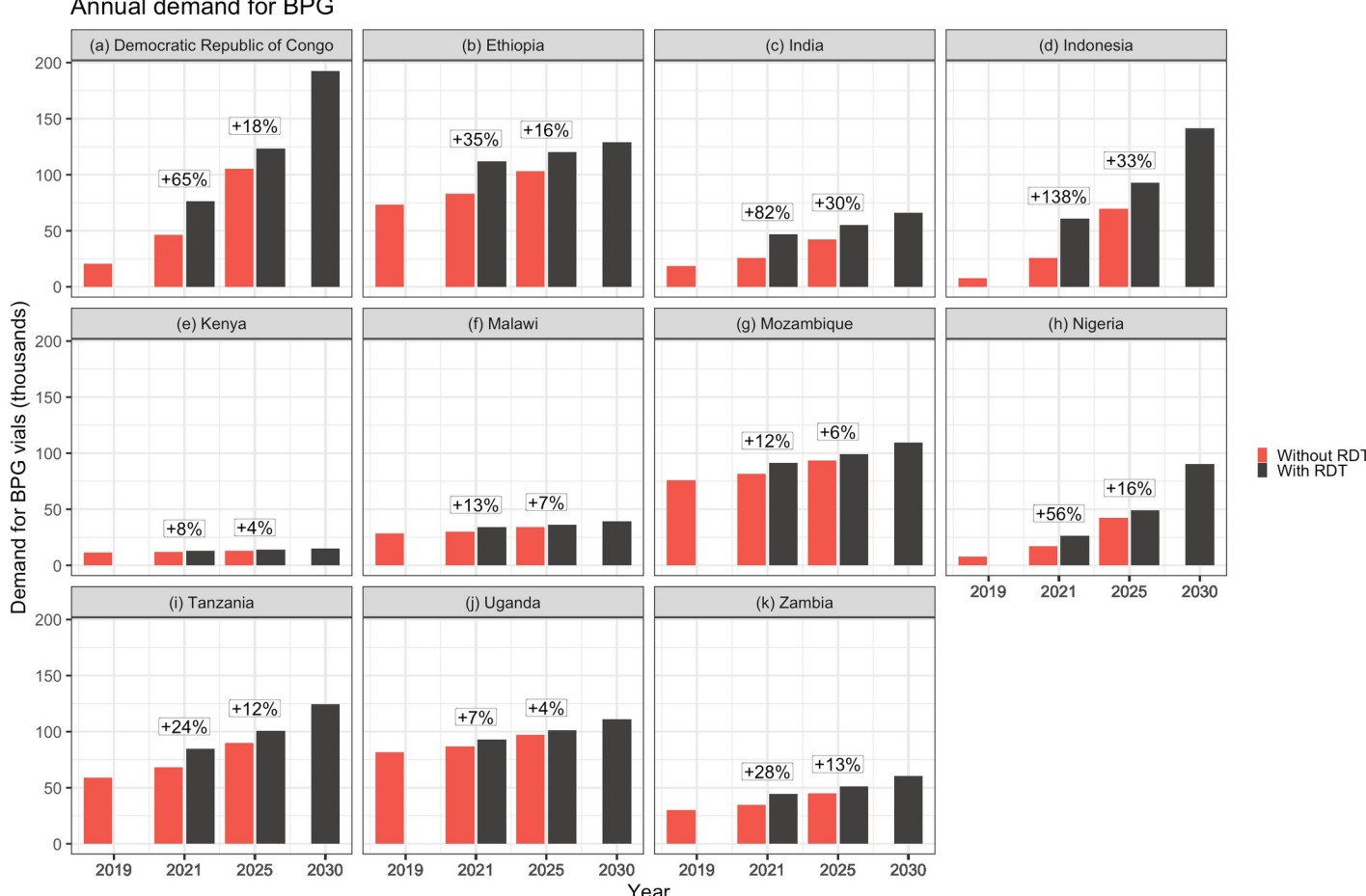

**Fig 1. Increase in demand of BPG for treating syphilis positive PW in 11 focus countries.** Graph indicates the quantity of 2.4 MU BPG vials required for treatment of PW identified syphilis positive with or without dual HIV/syphilis RDT if dual tests replace all syphilis testing in 2021.

Table 2. Demand for treatment regimen for PW.

| Country | Year | BPG for Pregnant Women[a] | | |
|---|---|---|---|---|
| | | (a) Without Dual RDT | (b) With Dual RDT | % Change[b] between (a) and (b) |
| Democratic Republic of Congo | 2019 | 20,343 | - | - |
| | 2021 | 46,236 | 76,506 | 65% |
| | 2025 | 105,159 | 123,570 | 18% |
| | 2030 | 192,369 | 192,369 | 0% |
| Ethiopia | 2019 | 73,443 | - | - |
| | 2021 | 83,181 | 112,146 | 35% |
| | 2025 | 103,326 | 120,243 | 16% |
| | 2030 | 128,958 | 128,958 | 0% |
| India | 2019 | 18,264 | - | - |
| | 2021 | 25,677 | 46,728 | 82% |
| | 2025 | 42,321 | 55,224 | 30% |
| | 2030 | 65,922 | 65,922 | 0% |
| Indonesia | 2019 | 7,926 | - | - |
| | 2021 | 25,620 | 61,059 | 138% |
| | 2025 | 69,792 | 92,886 | 33% |
| | 2030 | 141,516 | 141,516 | 0% |
| Kenya | 2019 | 11,298 | - | - |
| | 2021 | 11,880 | 12,813 | 8% |
| | 2025 | 13,152 | 13,692 | 4% |
| | 2030 | 14,928 | 14,928 | 0% |
| Malawi | 2019 | 28,206 | - | - |
| | 2021 | 30,099 | 33,993 | 13% |
| | 2025 | 34,041 | 36,354 | 7% |
| | 2030 | 39,297 | 39,297 | 0% |
| Mozambique | 2019 | 76,020 | - | - |
| | 2021 | 81,477 | 91,131 | 12% |
| | 2025 | 93,345 | 99,183 | 6% |
| | 2030 | 109,389 | 109,389 | 0% |
| Nigeria | 2019 | 7,794 | - | - |
| | 2021 | 16,851 | 26,316 | 56% |
| | 2025 | 42,132 | 48,900 | 16% |
| | 2030 | 90,273 | 90,273 | 0% |
| Tanzania | 2019 | 59,187 | - | - |
| | 2021 | 68,370 | 84,711 | 24% |
| | 2025 | 90,048 | 100,857 | 12% |
| | 2030 | 124,293 | 124,293 | 0% |
| Uganda | 2019 | 81,696 | - | - |
| | 2021 | 86,796 | 93,216 | 7% |
| | 2025 | 97,398 | 101,277 | 4% |
| | 2030 | 111,075 | 111,075 | 0% |
| Zambia | 2019 | 30,282 | - | - |
| | 2021 | 34,761 | 44,448 | 28% |
| | 2025 | 44,982 | 51,012 | 13% |
| | 2030 | 60,408 | 60,408 | 0% |

(Continued)

**Table 2.** (Continued)

| Country | Year | BPG for Pregnant Women[a] | | |
|---|---|---|---|---|
| | | (a) Without Dual RDT | (b) With Dual RDT | % Change[b] between (a) and (b) |
| Total | 2019 | 414,459 | - | - |
| | 2021 | 510,948 | 683,067 | 34% |
| | 2025 | 735,696 | 843,198 | 15% |
| | 2030 | 1,078,428 | 1,078,428 | 0% |

2.4MU vials of BPG, three vials per pregnant woman

[b] This indicates the difference in the demand for BPG between scenarios (a) and (b), specifically the incremental change in demand for BPG when dual tests are implemented in the screening process instead of two separate screening tests for syphilis and HIV.

Estimates of change in benzathine penicillin demand for treatment of PW with syphilis with the implementation of dual HIV/syphilis RDTs in ANC in 11 priority countries.

syphilis would simultaneously sync with screening coverage of HIV. In 2025, continued linear scaleup of rapid dual HIV/syphilis testing would increase the BPG demand to 735,696 vials (>100% over 2019 baseline). It was assumed that all countries will achieve WHO-defined EMTCT targets for syphilis (95% screening and treatment coverage) in 2030, with or without rapid dual testing. BPG demand was expected to grow to 1,078,428 vials annually in 2030, across these eleven focus countries (Table 2).

The countries projected to have the highest immediate increase (in 2021) in BPG demand upon adoption of dual HIV/syphilis RDTs are Indonesia (+35,439 vials or +138%), Democratic Republic of Congo (+30,270 vials or +65%), and Ethiopia (+28,965 vials or +35%). The countries with the smallest increase in demand are Kenya (+933 vials or +8%), Malawi (+3,894 vials or +13%), and Uganda (+6,420 vials or +7%), reflecting a smaller gap between reported HIV and syphilis screening coverage for these countries (Fig 1).

### Gradual rollout of dual tests–India snapshot

If dual testing is implemented gradually, the increase in demand for BPG vials would be lower initially in comparison to the scenario where HIV screening is entirely replaced by dual testing in 2021. BPG demand for treatment of PW in India was estimated with the assumption that dual testing is scaled-up linearly from 2021 to 2025 (Fig 2). In this scenario, while there will be no difference in BPG demand in 2021, it will increase by 35% in 2025 (Fig 2). There is a gradual rollout of dual tests starting in 2021 up to 2025 (modelled linearly for simplicity). In 2025, 100% of the HIV testing in the field is performed using dual tests, which brings the syphilis testing coverage at par with that for HIV. Beyond 2025, both HIV and syphilis coverage stay equal (concurrent lines), as both are done through simultaneous dual testing.

### Demand for penicillin regimens for treatment of infants

Using 2019 data as a baseline estimate, 15,755 infant penicillin treatment doses would be required for treatment of infants born to women diagnosed with syphilis but not treated (Table 3). Replacement of single HIV test with rapid dual HIV/syphilis testing of pregnant women in 2021 would result in a 76% demand increase for infant penicillin doses to 31,322 doses reflecting the surge in detection of pregnant women with syphilis prior to scale up of treatment coverage. With continued gradual scale up of both screening and treatment of PW with syphilis, the number of infant penicillin doses would decline to 24,751 in 2025 and to

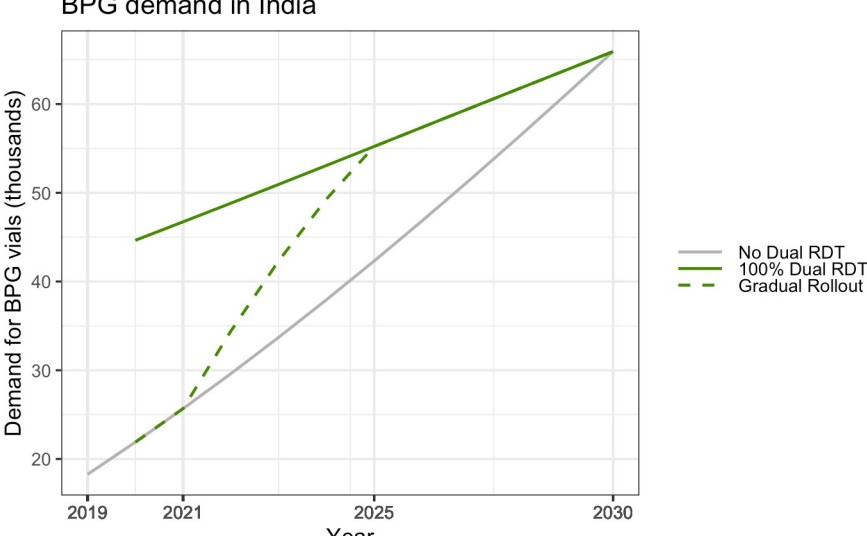

**Fig 2. BPG demand for treatment of PW in a phased dual HIV/syphilis RDT roll-out scenario (India snapshot).**
Graph indicates the demand of 2.4 MU BPG vials in India under 3 different cases–no access to dual tests, access to dual tests available to all PW, access to dual tests increases gradually from 2021 to 2025.

10,413 doses in 2030 when all focus countries are assumed to have achieved 95% coverage of these EMTCT syphilis service indicators. (Table 3).

## Discussion

Estimating BPG demand for treatment of maternal syphilis in 11 focus countries following adoption of the WHO-recommended rapid dual HIV/syphilis tests, demonstrated: (a) a generalized increase in demand for BPG across focus countries resulting from increased syphilis detection in PW following increases in screening coverage; (b) overall demand of 2.4 MU vials would more than double from 415,000 vials in 2019 in these countries to over 1 million vials by 2030; (c) immediate demand increase is higher in countries with a larger gap between HIV and syphilis screening; and (d) demand for infant treatment regimens would decline over time as country programs approach EMTCT targets of 95% syphilis screening and treatment among PW. These findings demonstrate the need for countries to anticipate and adequately forecast increases in BPG demand alongside scale up of rapid dual HIV/syphilis testing in ANC.

**Table 3. Demand for treatment regimens for infants.**

| Year | Penicillin Regimens for Infants[b] | | |
|---|---|---|---|
| | (a) Without Dual RDT | (b) With Dual RDT | % Change between (a) and (b) |
| 2019 | 15,755 | - | - |
| 2021 | 17,788 | 31,322 | 76% |
| 2025 | 19,910 | 24,751 | 24% |
| 2030 | 10,413 | 10,413 | 0% |

Demand for penicillin regimens for treatment following the implementation of maternal testing dual HIV/syphilis RDTs in ANC in 11 priority countries

b Figures indicate number of infants requiring treatment for congenital syphilis, with any of the WHO-recommended treatment regimens for infants

As of 2020, three dual HIV/syphilis RDT products have secured WHO prequalification, a global quality assurance standard [18]. The WHO recommendation [10] for use of these rapid dual tests during ANC will allow a matching percentage of pregnant women to be screened simultaneously for syphilis and HIV using the same fingerstick blood sample. Dual RDT scale-up efforts should be also undertaken in tandem with accurate procurement of BPG for treatment of syphilis reactive PW. Due to the underlying market risks to BPG supply globally, programs may also need better visibility into delivery lead times to prevent stockouts [14]. Countries can use the WHO Congenital Syphilis Estimation Tool [17] to generate national level estimates of BPG demand with scale up of maternal syphilis screening and treatment.

The incremental BPG demand depends on the rate of dual testing scale-up in the country. For example, the national HIV and STI policy in India recommends dual testing as the key element of the national strategy to eliminate congenital syphilis [19]. However, owing to the country's size, roll-out across the country would likely take place in a phased manner over 2 to 5 years, and BPG demand will increase gradually as opposed to a steep increase as in the case of immediate 100% replacement. In contrast, owing to improvement of maternal syphilis screening and treatment coverage over time with adoption of dual testing, fewer infants would be born with congenital syphilis, and hence demand for infant treatment doses would fall gradually.

There are limitations that should be considered when interpreting these analyses and estimates. The programmatic data for all indicators in all of the 11 focus countries could not be extracted for the same years due to gaps in data reporting. Uniform extrapolation of historical figures to present current status may diverge from the actual program data.

Additionally, the linear growth model adopted for modelling the scale-up of screening and treatment coverage for all focus countries may not accurately reflect the ground realities in the countries. Countries will differ in the pace of scale-up, and the gains may slow down as the countries approach 100% coverage. These realities have been ignored to allow for simplicity in modelling, and a linear growth towards 95% is assumed.

100% substitution of HIV tests to implement dual testing will most certainly not occur instantaneously in the focus countries. However, this has been assumed to account for the maximum potential syphilis screening coverage that can be achieved with dual testing. The separate analysis performed for India takes into account this ground reality and presents a more likely picture of gradual adoption of dual tests. The objective of this publication was to assist countries mitigate shortages of BPG observed globally in the past. With that aim in mind, the authors chose to present an upper bound of the requirement of BPG in the near future. The analysis, while aspirational in the near future, would reflect the reality as countries get closer to EMTCT targets in the latter half of the decade.

The analysis is based on the hypothetical scenario that all focus countries will achieve EMTCT in 2030, which may not be true for all settings. While different countries may vary in their pace to achieve EMTCT targets, the analysis was performed with the WHO goal of EMTCT by 2030 in mind. Hence, the demand for BPG evaluated in this analysis in turn indicates the number of vials country programs need to procure in order to fulfil 95% treatment coverage in PW with syphilis by 2030.

This study estimates BPG demand for maternal and congenital syphilis only. Other conditions requiring BPG such as syphilis in men and non-pregnant women, group A streptococcal infection, rheumatic heart disease, and yaws among others have not been considered.

The analysis uses BPG dosing of three 2.4 MU injections per PW with syphilis as recommended by WHO for treatment of syphilis of unknown duration or late latent stage infection. Those with infection in the primary, secondary and early latent stages may be treated with a single dose of BPG and WHO guidance considers a single dose of 2.4 MU of BPG given to the

pregnant woman with syphilis to be adequate for prevention of mother-to-child transmission [4]. Additionally, a recent review of global guidelines for treatment of maternal syphilis by Trinh et al [20] suggests that different countries have varied guidelines to treat syphilis in PW. While 8 of the 59 guidelines reviewed recommend a triple dose of 2.4 MU of BPG to be administered regardless of syphilis stage, an additional 45 of 59 country guidelines recommend stage-based administration of BPG. Only 6 countries surveyed recommended a single dose of 2.4 MU of BPG for all stages of infection. Screening tests like the dual HIV/Syphilis RDT only indicate the presence of syphilis and do not specify stage or duration of infection in a PW. While syphilis manifests and progresses in various ways, it is difficult to estimate stage or duration of infection based on the presence (or absence) of symptoms alone as a majority of infections can occur without detectable symptoms, and RPR titre-based testing is not feasible outside laboratory settings. Hence, a deliberate assumption of three doses has been made to estimate the BPG demand based on WHO recommendations for treatment of pregnant women with syphilis of unknown duration [4].

Thus, overestimation of actual demand may have occurred due to differing treatment practices. Actual demand may vary between a third of the evaluated demand and the evaluated demand from this analysis. In light of global shortages of BPG [14], the relatively low price of BPG vials, and the devastating effect of maternal and congenital syphilis, the risk of underestimation outweighs that of overestimation of demand. Country programs may modify the estimates as per national guidelines and through use of the WHO Congenital Syphilis Estimation Tool [17].

Use of rapid dual HIV/syphilis tests are not recommended for women known to be HIV-infected [12]. Thus, a separate syphilis test would be required in geographies with high HIV prevalence among PW. However, country programs are still expected to test and treat all PW for syphilis, regardless of their HIV status. The WHO 2030 EMTCT targets for maternal syphilis screening and treatment are agnostic of the type of test used, and BPG demand is not affected by the same. While HIV positive PW will not be tested for HIV (which is proposed to be replaced with dual testing), country programs are expected to screen HIV pregnant women for syphilis. These BPG estimates assume that scale up of syphilis screening would still occur where HIV prevalence among pregnant women precludes use of the rapid dual HIV/syphilis test.

Finally, authors have reservations regarding the countries reporting 100% treatment among PW with syphilis, owing to the known challenges in effective delivery and administration of BPG. Moreover, though 100% treatment coverage implies that no infants would require treatment for syphilis, countries should be prepared to procure infant treatment dosages of penicillin as programs may not have mechanisms to ascertain whether the treatment of the mothers was timely and adequate.

## Conclusions

Implementation of dual HIV/syphilis RDTs will increase access to ANC syphilis screening and diagnosis. Women and infants in countries with high gaps in HIV and syphilis screening would benefit on a larger scale with adoption of dual HIV/syphilis testing. Implementing dual testing in antenatal care programs could also contribute to achieving dual EMTCT goals earlier than projected, particularly in those countries which have already made significant progress towards EMTCT of HIV. As country programs inch closer to EMTCT targets, diagnosing and treating more PW with syphilis would accelerate the elimination of congenital syphilis.

This publication aims to assist country programs in forecasting BPG demand for treatment of antenatal syphilis. Prioritization of universal access to BPG for congenial syphilis

prevention—which remains the second leading cause of stillbirth globally–calls for adequate resourcing, demand forecasting, sufficient procurement, appropriate distribution to ANC clinics, and training of healthcare providers in the timely maternal administration of this life-saving medication.

## Appendix

### Additional methods

**Rationale behind choice of geographies.** We selected geographies which represent the largest estimates of PW and/or the highest gaps in HIV and syphilis screening coverage for the analysis. 80% of the total worldwide pregnancies can be attributed to 13 countries–India, China, Nigeria, Indonesia, Brazil, Bangladesh, Ethiopia, Democratic Republic of Congo, Mexico, Tanzania, Kenya, Uganda, and South Africa [21]. An earlier study also identified 9 countries with recorded differences between HIV and syphilis screening coverage—Ethiopia, India, Indonesia, Kenya, Malawi, Mozambique, Nigeria, Uganda and Zambia [13]. Combining these two data sets includes 16 countries. Five countries were removed from the analysis as HIV or syphilis screening coverage was unavailable in recently published databases–Bangladesh, Brazil, China, Mexico and South Africa. The study was performed on the HIV and syphilis screening and treatment data reported by the remaining 11 countries to estimate the BPG demand. These 11 high-burden countries represented 37% of the world's pregnancies and 38% of the world's congenital syphilis positive births in 2016 [7].

**Congenital syphilis estimation tool.** In 2017, the WHO developed the Congenital Syphilis Estimation Tool [17] for countries to estimate current and historical congenital syphilis case numbers, case rates and adverse birth outcomes (ABOs) and to look at future trends in CS cases under different scenarios of ANC coverage and maternal syphilis testing and treatment. We used methods from this Tool to estimate case rates and hence, dosages of BPG required to treat these maternal syphilis cases. The target coverage for EMTCT programs was considered as 95% for the three metrics–(a) PW attending ANC at least once, (b) PW screened for syphilis out of those registered for ANC, and (c) PW treated with BPG out of those found syphilis reactive. The target year for the present study was 2030 for all focus countries consistent with the global target year set for congenital syphilis elimination in the WHO Global Health Sector Strategy for Sexually Transmitted Infections [22].

**Data sources.** The number of live births as extracted from the UNDP World Population Prospects 2015 was considered a proxy for the number of pregnancies every year in the 11 focus countries. Note that the actual number of pregnancies may vary across geographies due to different levels of foetal loss and stillbirths. The data for ANC coverage for the years 2012 and 2016 were obtained from Korenromp et al and were linearly extrapolated till 2019. The HIV screening coverage for the years 2015 to 2019 were extracted from UNAIDS's report for HIV testing during ANC where available (UNAIDS, 2020). Syphilis screening, positivity and treatment figures for 2019 were derived from the WHO Global Health Observatory (GHO) data repository [15]. For those countries where syphilis screening, positivity or treatment data were unavailable for 2019, a trend analysis was conducted to extrapolate data for 2019 using the most recent year data available.

The data for HIV and syphilis screening coverage has been extracted from different sources, which may lead to variation in their denominators. While UNAIDS considers population-based pregnancy estimates as denominator for HIV screening coverage, GHO defines syphilis screening coverage out of PW attending ANC [15, 16]. As the estimate for PW attending ANC may be less than 100%, the gap between the actual screening coverage for HIV and syphilis might be larger than that used in this analysis resulting in underestimation.

**Extrapolation of screening and treatment coverage.** For ANC coverage, if a country showed an increase in coverage between 2012 and 2016, a linear model was projected to 2019 with the ceiling value fixed at 100% coverage. If a country showed a decrease or stagnancy in ANC coverage between 2012 and 2016, the 2016 values were retained for every year until 2019. This assumed that the countries showing an improvement in ANC coverage would continue the same trend until they reach the target 95%, and those showing a decrease in coverage would, at the very least, arrest the decline and continue to consistently perform at the levels demonstrated in 2016. This approach was replicated for syphilis screening and treatment coverage wherever the data were unavailable from the GHO repository. For countries where syphilis positivity data were unavailable from GHO data, prevalence estimates from WHO estimates [7] for the year 2016 were considered as the proxy for syphilis positivity among PW.

For HIV screening coverage, the data reported by UNAIDS was used for analysis and the countries for which HIV screening figures were not available for 2019, they were calculated as a linearly weighted increase for the previous four years, as below.

$$HIV_{2019} = HIV_{2018}$$
$$+ (3 \times (HIV_{2018} - HIV_{2017}) + 2 \times (HIV_{2017} - HIV_{2016}) + 1 \times (HIV_{2016}$$
$$- HIV_{2015}))/6$$

Because 2016 UNAIDS data are missing, HIV screening coverage in 2016 was taken as an average for 2015 and 2017. If the formula yielded an HIV screening coverage lower than 2018, it was assumed that the decline would stagnate, and screening coverage would remain consistent at figures reported in 2018.

Beyond 2019, the ANC1 coverage, HIV screening coverage, syphilis screening coverage and syphilis treatment coverage was extrapolated linearly to EMTCT targets (95%) in 2030. This was done to model the focus countries' efforts to scaleup screening and treatment for HIV and syphilis in line with the target to achieve EMTCT by 2030. The ceiling for all coverage figures was fixed at 100%.

The maternal syphilis prevalence data were assumed to be constant over time.

**Calculating demand for BPG.** ANC coverage is the proportion of PW visiting an ANC facility at least once during their pregnancy.

The treatment required for maternal syphilis infection varies according to syphilis stage, from one to three doses of intramuscular BPG (2.4 MU) [4]. However, with limited access to data indicating number of doses administered, it was assumed that each PW testing positive for syphilis would be treated with three 2.4 MU doses of BPG for prevention of congenital syphilis.

Confirmed or suspected cases of congenital syphilis due to inadequate or no treatment of pregnant women with syphilis should be treated with penicillin G or procaine penicillin G [4].

## Supporting information

**S1 File. BPG forecasting analysis (2019–30).** An analysis of BPG demand for treatment of maternal and congenital syphilis in 11 focus countries (Excel Workbook).
(XLSM)

**S1 Data. BPG estimation analysis.**
(XLSM)

## Acknowledgments

**Disclaimer:** The views expressed in this manuscript are those of the authors and do not necessarily represent the official position of the World Health Organization, the United States of America Centers for Disease Control and Prevention or Clinton Health Access Initiative.

## Author Contributions

**Conceptualization:** Sapan Shah, Surbhi Garg, Obiageli Ofili, Yashika Bansal, Frederic Seghers, Andrew Storey, Melanie Taylor.

**Data curation:** Sapan Shah, Katherine Heath, Melanie Taylor.

**Formal analysis:** Sapan Shah, Melanie Taylor.

**Funding acquisition:** Obiageli Ofili, Yashika Bansal, Andrew Storey, Melanie Taylor.

**Investigation:** Sapan Shah.

**Methodology:** Sapan Shah, Melanie Taylor.

**Project administration:** Obiageli Ofili, Yashika Bansal, Melanie Taylor.

**Resources:** Katherine Heath, Obiageli Ofili, Yashika Bansal, Melanie Taylor.

**Supervision:** Surbhi Garg, Katherine Heath, Obiageli Ofili.

**Validation:** Sapan Shah, Surbhi Garg, Obiageli Ofili, Yashika Bansal, Frederic Seghers, Andrew Storey, Melanie Taylor.

**Visualization:** Sapan Shah, Katherine Heath, Melanie Taylor.

**Writing – original draft:** Sapan Shah, Surbhi Garg, Melanie Taylor.

**Writing – review & editing:** Sapan Shah, Surbhi Garg, Katherine Heath, Obiageli Ofili, Yashika Bansal, Frederic Seghers, Andrew Storey, Melanie Taylor.

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
