## [Decision Letter · Decision Letter 0]

2 Jun 2021

PONE-D-21-12082

Estimation of benzathine penicillin G demand for congenital syphilis elimination with adoption of dual HIV/syphilis rapid diagnostic tests in eleven high burden countries

PLOS ONE

Dear Dr. Shah,

Thank you for submitting your manuscript to PLOS ONE. After careful consideration, we feel that it has merit but does not fully meet PLOS ONE’s publication criteria as it currently stands. Therefore, we invite you to submit a revised version of the manuscript that addresses the points raised during the review process.

Be sure to fully address the reviewers comments to make this manuscript as clear as possible and useful for policy makers moving forward. 

We look forward to receiving your revised manuscript.

Kind regards,

Julie AE Nelson, PhD

Academic Editor

PLOS ONE

Journal Requirements:

Reviewers' comments:

Reviewer's Responses to Questions

**Comments to the Author**

1. Is the manuscript technically sound, and do the data support the conclusions?

Reviewer #1: Yes

Reviewer #2: Yes

2. Has the statistical analysis been performed appropriately and rigorously? 

Reviewer #1: Yes

Reviewer #2: N/A

3. Have the authors made all data underlying the findings in their manuscript fully available?

Reviewer #1: Yes

Reviewer #2: Yes

4. Is the manuscript presented in an intelligible fashion and written in standard English?

Reviewer #1: Yes

Reviewer #2: Yes

5. Review Comments to the Author

Reviewer #1: In this analysis, projections for scale up of syphilis screening in ANC is presented in 11 countries. The authors highlight the need plan for potential increases in treatment with BPG if scale-up is to occur. While results are somewhat intuitive (as you scale-up testing coverage you’ll need more treatment) it may be helpful for countries to plan for treatment scale-up needs and to quantify this. However, some of the assumptions (3 doses of BPG) and 100% switch in testing type do not seem realistic, presenting alternative scenarios that are more realistic may help programs more appropriately plan for the magnitude of doses they may actually require. However, the authors do acknowledge the WHO tool is available for countries to make their own projections.

Abstract

– Include some details how you derived these results; main body text says “extrapolated” – does this mean modeling?

Methods

Can you be more specific on extrapolating? Was this a linear increase from current coverage to 95% evenly distributed over 10 years?

Did you consider contraindications for the dual test for scenario b (mentioned in discussion)? It is unlikely to be 100% as there are a large % of WLWH in many countries. Considering conducting some sensitivity analysis around this would be helpful.

Can you be more specific about how you derived number who would be positive? Summarize methods from the WHO tool and how this is derived? Ie, what are the inputs? Some details here in main text would be helpful; not just the supplementary material.

The assumption that all women would get 3 doses seems unlikely; did the authors consider an average based on what women typically get?

The assumption that tests are swapped out in 2021 seems highly unlikely, can the authors justify this decision rather than a more realistic scenario? The linear scale-up used for India seems more appropriate.

The figure showing India’s rollout of Dual RDT is hard to interpret. Is the assumption that between 2020-2025 there is gradual rollout but then it becomes 100% in 2025? Or are the lines overlapping.

How was the slope of rollout determined for India?

It may help to subtract out the baseline demand in addition to absolute to more clearly demonstrate the difference attributed to testing changes.

Table 1. Please clarify % change is from the prior timepoint. Also, this is a little deceptive as % change from 2019-2021 is only 2 years while 2021-2025 is 4 years. Why not present the same interval increments for more transparency about change over time?

The limitation that all countries will achieve EMTCT in 2030 is good to acknowledge. Can the authors justify their decision for this assumption rather than perhaps a more realistic assumption?

The authors should also acknowledge assumptions about testing coverage and pace of change of tests in the limitations.

Reviewer #2: Summary of research and overall impression

This study highlights the potential need for increasing quantities of benzathine pencillin G (BPG) with increasing syphilis testing coverage, particularly in light of likely scale up of the dual HIV-syphilis rapid diagnostic test. The topic is important, as BPG supply has previously hampered ability to successfully treat maternal syphilis, and advance projections are needed to ensure adequate quantities for maternal treatment and EMTCT of syphilis.

Overall, the authors have clearly articulated the purpose, results, and conclusions of the article. Additional clarification of the methods/scenarios in the main paper would help make the results easier to interpret.

Discussion of specific areas for improvement

1. Recommend adding additional content to the methods section of the main paper so that you can understand some of the basic assumptions of the calculations without having to read the additional methods (provide summary statement in methods then reference additional methods for further detail).

a. Line 117, Calculation for estimating demand for BPG: It would be helpful below the calculation to summarize which variables are modified in the scenarios and which are considered constant over time from 2019 to 2030 (or modeled in a different way). From additional methods, I gather:

i. ANC coverage constant over time from 2019-2030? (after 2019 calculation in additional methods)

ii. HIV testing coverage constant over time from 2019-2030? (after 2019 calculations in additional methods)

iii. Syphilis positivity – constant over time

b. Line 124, Estimating demand for treatment regimens for infants

i. After calculation, it would be helpful to have some summary sentence on how these variables are determined briefly (e.g. pregnant women provided with BPG scaled up to 95% coverage by 2030), then reference to additional methods.

2. Clarify the scenarios starting on line 131:

a. Was ANC coverage scaled up from 2019 to 2030 to reach the 95% target for EMTCT? I believe it was kept constant, but if all other targets were scaled to reach EMTCT by 2030, please explain why did you not scale ANC coverage to reach 95%.

b. Consider re-naming scenario (a) since maintaining the “status quo” would not get these countries to EMTCT targets by 2030, so even scenario (a) assumes some additional investment/focus on improving syphilis testing and treatment coverage, but not rapid implementation of the dual test.

c. For scenario (b) description, since the scenario continues to 2030, please add what will happen in this scenario after immediate adoption of the dual test (e.g. increased to level of HIV screening coverage immediately (in 2021), then linear progress to EMTCT by 2030?)

3. Consider adding additional limitations based on calculation assumptions, particularly for ANC coverage – if countries with low ANC coverage (e.g. Nigeria, DRC, Ethiopia) increase ANC coverage to reach EMTCT targets, this could cause changes in projections.

4. Additional methods line 378: ANC coverage estimates – it seems more appropriate to use the UNICEF dataset for ANC1 coverage – includes many updates beyond 2016. Please consider using this or clarify why it was not the preferred source for ANC coverage data.

Minor issues:

1. Abstract: Conclusions “country programs will need to synergize with disease surveillance programs” – Recommend clarifying since clinical and disease surveillance are both country programs – I believe you mean MNCH clinical programs and disease surveillance programs in country.

2. It would provide helpful context to include a baseline data table for the 11 focus countries either in main paper or additional materials – e.g. show baseline ANC coverage, HIV testing coverage, syphilis testing coverage, syphilis positivity, syphilis treatment coverage. Since country scenarios vary based on where they start with these indicators, it would put the changes over time in perspective. Additionally, in Line 253 of Discussion, you mention countries with 100% treatment among PW with syphilis but this baseline data is not shown anywhere, so would be helpful to have all data presented in discussion also available in results.

3. Table 1 and 2: Recommend changing column title from “% change” to “% difference between scenario A and B” or something similar. In my initial reading, I assumed % change referred to change over time (e.g. between 2019 and 2021) – since the two scenarios are options, and you would not change between one and the other, it may be clearer to call this a difference (in table and text).

4. Line 248 paragraph: Consider rewording this section on issues with syphilis testing for HIV-positive women. Describing places “where HIV prevalence precludes use of .. dual test” seems to indicate that dual test would not be appropriate for high HIV prevalence settings (like many of the countries included here), but this should not be the case. Instead, I think you assume syphilis screening would still occur among all women, including HIV-positive women who would need a separate syphilis test.

6. PLOS authors have the option to publish the peer review history of their article (what does this mean?). If published, this will include your full peer review and any attached files.

Reviewer #1: No

Reviewer #2: No

---

## [Author Response · Author response to Decision Letter 0]

24 Jun 2021

"Response to Reviewers.docx" is included in the uploaded files. Given below is a plain text translation of the file.

Manuscript reference number: PONE-D-21-12082

Title: Estimation of benzathine penicillin G demand for congenital syphilis elimination with adoption of dual HIV/syphilis rapid diagnostic tests in eleven high burden countries

Dear Editors/Reviewers,

Thank you for your careful review of our manuscript entitled “Estimation of benzathine penicillin G demand for congenital syphilis elimination with adoption of dual HIV/syphilis rapid diagnostic tests in eleven high burden countries” (PONE-D-21-12082).

We greatly appreciate your comments and have revised the manuscript and have detailed responses to each of the points below. Original reviewer comments are included in italics. Investigators’ responses to reviewers are included in bold. Revised text is included in bold italics. 

Journal Requirements:

Please review your reference list to ensure that it is complete and correct. If you have cited papers that have been retracted, please include the rationale for doing so in the manuscript text or remove these references and replace them with relevant current references. Any changes to the reference list should be mentioned in the rebuttal letter that accompanies your revised manuscript. If you need to cite a retracted article, indicate the article’s retracted status in the References list and also include a citation and full reference for the retraction notice.

We carefully checked the PLOS ONE’s referencing style requirements and have ensured that the manuscript fulfils all criteria. To the best of our knowledge, the references list is complete and current.

In order to address some of the reviewers’ comments, we have cited a recent publication to the Sexual and Reproductive Health Matters journal and added it as the 22nd item to the references list as below:

Trinh T, Leal AF, Mello MB, Taylor MM, Barrow R, Wi TE, Kamb ML. Syphilis management in pregnancy: a review of guideline recommendations from countries around the world. Sex Reprod Health Matters. 2019 Dec;27(1):69-82. 

We carefully checked the PLOS ONE’s style requirements and have revised our manuscript as well as file names accordingly.

Please include captions for your Supporting Information files at the end of your manuscript, and update any in-text citations to match accordingly.

The necessary caption for the supporting information file has been placed in the appendix as per the reviewer’s suggestion and PLOS ONE style requirements.

Reviewer #1: 

In this analysis, projections for scale up of syphilis screening in ANC is presented in 11 countries. The authors highlight the need plan for potential increases in treatment with BPG if scale-up is to occur. While results are somewhat intuitive (as you scale-up testing coverage you’ll need more treatment) it may be helpful for countries to plan for treatment scale-up needs and to quantify this. However, some of the assumptions (3 doses of BPG) and 100% switch in testing type do not seem realistic, presenting alternative scenarios that are more realistic may help programs more appropriately plan for the magnitude of doses they may actually require. However, the authors do acknowledge the WHO tool is available for countries to make their own projections.

We thank the reviewer for their comments and suggestions. Detailed responses to individual points called out above are appended below with the specific comments.

Abstract

Include some details how you derived these results; main body text says “extrapolated” – does this mean modeling?

The program coverage figures (ANC1, syphilis screening, HIV screening and syphilis treatment) were modelled to linearly increase from their values in 2019 to 95% (EMTCT targets) in 2030. The Methods section in the abstract has been revised as per the reviewer’s suggestion, adding some more detail as below:

Country-reported ANC coverage, PW syphilis screening and treatment coverage values in 2019 were scaled linearly to EMTCT targets by 2030 (constant increasing slope from 2019 figures to 95% in 2030) for 11 focus countries.

Methods

Can you be more specific on extrapolating? Was this a linear increase from current coverage to 95% evenly distributed over 10 years?

Refer to response above.

Did you consider contraindications for the dual test for scenario b (mentioned in discussion)? It is unlikely to be 100% as there are a large % of WLWH in many countries. Considering conducting some sensitivity analysis around this would be helpful.

We agree with the reviewer’s observation that dual testing may not be used for 100% of the PW owing to a fraction of them already knowing their HIV status, and have mentioned as much in our Discussion section. However, country programs are still expected to test and treat all PW for syphilis, regardless of their HIV status. Syphilis screening and treatment strategy is agnostic of the type of test used, and BPG demand is not affected by the same. While WLWH will not be tested for HIV (which is proposed to be replaced with dual testing), country programs are expected to screen all HIV positive pregnant women for syphilis.

Can you be more specific about how you derived number who would be positive? Summarize methods from the WHO tool and how this is derived? Ie, what are the inputs? Some details here in main text would be helpful; not just the supplementary material.

We agree with the reviewer’s suggestions on briefly including the calculations for determining the demand for BPG. We have added the following text after the formula to estimate BPG demand, in the revised manuscript:

Number of live births, reported by the United Nations Population Division, is used as an proxy for number of pregnancies in the WHO’s Congenital Syphilis Elimination Tool (17). ANC coverage, syphilis and HIV screening coverage and syphilis treatment coverage are included from the WHO Global Health Observatory which is populated with country-reported data. These values are evaluated as per the linear growth model described in the Methods. Maternal syphilis positivity rate is assumed to be constant at the levels recorded in 2019. Number of doses per PW is also kept constant at three doses per PW, owing to limited access to data indicating number of doses administered in the field. The values are then multiplied according to the formula above to evaluate the demand for BPG in any given year.

The assumption that all women would get 3 doses seems unlikely; did the authors consider an average based on what women typically get?

The reviewer’s observation is accurate. The assumption that all women would get all 3 doses is certainly unlikely. The treatment required for maternal syphilis infection varies according to syphilis stage, from one to three doses of BPG. However, with limited access to data indicating number of doses administered, it was assumed that each PW testing positive for syphilis would be treated with three 2.4 MU doses of BPG for prevention of congenital syphilis. Even in the authors’ home country India, the administration and dosage of BPG varies significantly across geographies, with only anecdotal evidence available for most regions. In order to simplify the analysis and present an upper bound of the requirement of BPG, the 3-dose assumption was taken.

The following text has been added to the Discussion section under the paragraph about the 3-dose assumption:

Additionally, a recent review of global guidelines for treatment of maternal syphilis by Trinh et al (22) suggests that different countries have varied guidelines to treat syphilis in PW. While 8 of the 59 guidelines reviewed recommend a triple dose of 2.4 MU of BPG to be administered regardless of syphilis stage, an additional 45 of 59 country guidelines recommend stage-based administration of BPG. Only 6 countries surveyed recommended a single dose of 2.4 MU of BPG for all stages of infection. Screening tests like the dual HIV/Syphilis RDT only indicate the presence of syphilis and do not specify stage or duration of infection in a PW. While syphilis manifests and progresses in various ways, it is difficult to estimate stage or duration of infection based on the presence (or absence) of symptoms alone as a majority of infections can occur without detectable symptoms, and RPR titre-based testing is not feasible outside laboratory settings. Hence, a deliberate assumption of three doses has been made to estimate the BPG demand based on WHO recommendations for treatment of pregnant women with syphilis of unknown duration (4).

Thus, overestimation of actual demand may have occurred due to differing treatment practices. Actual demand may vary between a third of the evaluated demand and the evaluated demand from this analysis. In light of global shortages of BPG (14),the relatively low price of BPG vials, and the devastating effect of maternal and congenital syphilis, the risk of underestimation outweighs that of overestimation of demand. Country programs may modify the estimates as per national guidelines and through use of the WHO Congenital Syphilis Estimation Tool (17).

The assumption that tests are swapped out in 2021 seems highly unlikely, can the authors justify this decision rather than a more realistic scenario? The linear scale-up used for India seems more appropriate.

We agree with the reviewer’s observation about the low likelihood of 100% swapping of HIV tests with dual tests in 2021. While some countries may be able to manage a quick turnaround, the same cannot be predicted or modelled for all focus geographies in this analysis. However, the objective of this publication was to assist countries in preventing and mitigating shortages of BPG observed globally in the past. With that aim in mind, the authors chose to present an upper bound of the requirement of BPG in the near future. The analysis, while aspirational in the near future, would reflect the reality as countries get closer to EMTCT targets in the later half of the decade. Additionally, the analysis for India was performed separately to account for this very reality of gradual scaleup, as the authors had directly contributed to the adoption of dual tests in the country.

The figure showing India’s rollout of Dual RDT is hard to interpret. Is the assumption that between 2020-2025 there is gradual rollout but then it becomes 100% in 2025? Or are the lines overlapping.

The reviewer’s observation is accurate. There is a gradual rollout of dual tests starting in 2021 up to 2025 (modelled linearly for simplicity). In 2025, 100% of the HIV testing in the field is performed using dual tests, which brings the syphilis testing coverage at par with that for HIV. Beyond 2025, both HIV and syphilis coverage stay equal (concurrent lines), as both are done through simultaneous dual testing. The following text has been added to the Results section of the manuscript:

There is a gradual rollout of dual tests starting in 2021 up to 2025 (modelled linearly for simplicity). In 2025, 100% of the HIV testing in the field is performed using dual tests, which brings the syphilis testing coverage at par with that for HIV. Beyond 2025, both HIV and syphilis coverage stay equal (concurrent lines), as both are done through simultaneous dual testing.

How was the slope of rollout determined for India?

As described above, the slope of dual testing roll-out in India was assumed to be linear. As the lead authors have contributed to the adoption of dual testing in India, they have been privy to the gradual, state-wise adoption of dual tests. While the efforts to implement dual testing in India began in early 2019, only 5 states (of a total of 29) had implemented dual testing in the field by the end of 2020. These 5 states accounted for about 20% of all pregnant women in the country. Assuming that the early adoption in select states would catalyse a speedup of implementation in others, the authors concluded that 2025 would be a realistic timeline for 100% adoption of dual testing. For simplicity in modelling of this analysis, the slope was assumed to be linear.

It may help to subtract out the baseline demand in addition to absolute to more clearly demonstrate the difference attributed to testing changes.

While the baseline shows the level of demand in 2019, it is not expected to stay stagnant over the years. Even without the implementation of dual testing, country programs are expected to scale up syphilis testing among PW to achieve EMTCT targets. This is assumed to be scenario (a) as explained above, and scenario (b) reflects the % increment over this. While the baseline figures are indicative, the authors have chosen not to subtract the same from the demand for every year to represent the absolute demand for BPG, as procurement will need to be done for the entire PW cohort, and not just the additional PW found reactive due to scale-up of testing.

Table 1. Please clarify % change is from the prior timepoint. Also, this is a little deceptive as % change from 2019-2021 is only 2 years while 2021-2025 is 4 years. Why not present the same interval increments for more transparency about change over time?

The % change is not defined from the previous timepoint. The change reflects the difference between the demand as dictated by scenario (a) vs that in scenario (b). The table headings have been revised to reflect the same.

The limitation that all countries will achieve EMTCT in 2030 is good to acknowledge. Can the authors justify their decision for this assumption rather than perhaps a more realistic assumption?

We agree with the reviewer’s comment. The following text has been added in the revised manuscript in the discussion section outlining the limitations of the analysis:

The analysis is based on the hypothetical scenario that all focus countries will achieve EMTCT in 2030, which may not be true for all settings. While different countries may vary in their pace to achieve EMTCT targets, the analysis was performed with the WHO goal of EMTCT by 2030 in mind. Hence, the demand for BPG evaluated in this analysis in turn indicates the number of vials country programs need to procure in order to fulfil 95% syphilis treatment coverage in PW by 2030.

The authors should also acknowledge assumptions about testing coverage and pace of change of tests in the limitations.

We agree with the reviewer’s comment. The following text has been added in the revised manuscript in the discussion section outlining the limitations of the analysis:

Additionally, the linear growth model adopted for modelling the scale-up of screening and treatment coverage for all focus countries may not accurately reflect the ground realities in the countries. Countries will differ in the pace of scale-up, and the gains may slow down as the countries approach 100% coverage. These realities have been ignored to allow for simplicity in modelling, and a linear growth towards 95% is assumed.

100% substitution of HIV tests to implement dual testing will most certainly not occur instantaneously in the focus countries. However, this has been assumed to account for the maximum potential syphilis screening coverage that can be achieved with dual testing. The separate analysis performed for India takes into account this ground reality and presents a more likely picture of gradual adoption of dual tests. The objective of this publication was to assist countries mitigate shortages of BPG observed globally in the past. With that aim in mind, the authors chose to present an upper bound of the requirement of BPG in the near future. The analysis, while aspirational in the near future, would reflect the reality as countries get closer to EMTCT targets in the latter half of the decade.

 

Reviewer #2: 

Summary of research and overall impression

This study highlights the potential need for increasing quantities of benzathine pencillin G (BPG) with increasing syphilis testing coverage, particularly in light of likely scale up of the dual HIV-syphilis rapid diagnostic test. The topic is important, as BPG supply has previously hampered ability to successfully treat maternal syphilis, and advance projections are needed to ensure adequate quantities for maternal treatment and EMTCT of syphilis.

Overall, the authors have clearly articulated the purpose, results, and conclusions of the article. Additional clarification of the methods/scenarios in the main paper would help make the results easier to interpret.

Discussion of specific areas for improvement

1. Recommend adding additional content to the methods section of the main paper so that you can understand some of the basic assumptions of the calculations without having to read the additional methods (provide summary statement in methods then reference additional methods for further detail).

We agree with the reviewer’s suggestions and have added summarizing text in the Methods section in the revised manuscript. Specific added sections are highlighted in the responses to questions posed below.

a. Line 117, Calculation for estimating demand for BPG: It would be helpful below the calculation to summarize which variables are modified in the scenarios and which are considered constant over time from 2019 to 2030 (or modeled in a different way). From additional methods, I gather:

i. ANC coverage constant over time from 2019-2030? (after 2019 calculation in additional methods)

ii. HIV testing coverage constant over time from 2019-2030? (after 2019 calculations in additional methods)

iii. Syphilis positivity – constant over time

We agree with the reviewer’s suggestion to summarize the evolution of variables with time. While the assumption (iii) that syphilis positivity is constant over time is correct, For (i) and (ii) the ANC1 coverage, syphilis and HIV testing coverage figures were also scaled linearly from 2019 to 2030, in line with country programs’ efforts to achieve WHO EMTCT targets of 95% coverage of of ANC1, 95% coverage of HIV and syphilis screening and 95% coverage of maternal syphilis (and HIV) treatment by 2030. However, a ceiling was fixed at 100%. This has been clarified by adding revised text in the Additional Methods section as below:

Beyond 2019, the ANC1 coverage, HIV screening coverage, syphilis screening coverage and syphilis treatment coverage was extrapolated linearly to EMTCT targets (95%) in 2030. This was done to model the focus countries’ efforts to scaleup screening and treatment for HIV and syphilis in line with the target to achieve EMTCT by 2030. The ceiling for all coverage figures was fixed at 100%.

The maternal syphilis prevalence data were assumed to be constant over time.

b. Line 124, Estimating demand for treatment regimens for infants

i. After calculation, it would be helpful to have some summary sentence on how these variables are determined briefly (e.g. pregnant women provided with BPG scaled up to 95% coverage by 2030), then reference to additional methods.

We agree with the reviewer’s suggestion on elaborating how the represented variables are evaluated. We added the following text after the equation to determine number of infant treatment regimens required:

The number of livebirths to syphilis positive PW is evaluated by multiplying syphilis prevalence, ANC1 coverage and syphilis screening coverage to total livebirths. The number of PW treated is calculated by multiplying this number with the syphilis treatment coverage.

2. Clarify the scenarios starting on line 131:

a. Was ANC coverage scaled up from 2019 to 2030 to reach the 95% target for EMTCT? I believe it was kept constant, but if all other targets were scaled to reach EMTCT by 2030, please explain why did you not scale ANC coverage to reach 95%.

ANC coverage was, in fact, scaled up to 95% in 2030. We agree with the reviewer’s observation that the language may not have conveyed the same explicitly. We have revised the description of scenario (a) as below:

ANC coverage, syphilis screening coverage and syphilis treatment scaled up linearly from 2019 to the target coverage 95% in the target year 2030 regardless of HIV screening coverage

b. Consider re-naming scenario (a) since maintaining the “status quo” would not get these countries to EMTCT targets by 2030, so even scenario (a) assumes some additional investment/focus on improving syphilis testing and treatment coverage, but not rapid implementation of the dual test.

We agree with the reviewer’s comment on the naming of scenario (a). We have replaced the name of scenario (a) from the manuscript with the following text:

(a) scaleup to EMTCT targets without dual testing

c. For scenario (b) description, since the scenario continues to 2030, please add what will happen in this scenario after immediate adoption of the dual test (e.g. increased to level of HIV screening coverage immediately (in 2021), then linear progress to EMTCT by 2030?)

We have added the following text as an addendum to the description of scenario (b), as suggested by the reviewer:

and then increased linearly to reach EMTCT targets by 2030

3. Consider adding additional limitations based on calculation assumptions, particularly for ANC coverage – if countries with low ANC coverage (e.g. Nigeria, DRC, Ethiopia) increase ANC coverage to reach EMTCT targets, this could cause changes in projections.

ANC1 coverage is also modelled to increase linearly from the levels in 2019 to EMTCT targets, i.e. 95% in 2030. The estimates of BPG demand account for this change. 

4. Additional methods line 378: ANC coverage estimates – it seems more appropriate to use the UNICEF dataset for ANC1 coverage – includes many updates beyond 2016. Please consider using this or clarify why it was not the preferred source for ANC coverage data.

While we agree with the reviewer’s comment about the UNICEF dataset containing updated data for ANC1 coverage beyond 2016, we chose not to include the data because of incompleteness of the dataset. Of the 11 focus countries chosen for this publication, the UNICEF dataset only contained ANC1 coverage data for 9. Of the 9, only 5 countries reported ANC1 coverage for 2016, 4 countries for 2017, 3 countries for 2018 and 2 countries for 2019. However, we performed a ‘sanity check’ of the extrapolated data from the present analysis comparing the data obtained from extrapolation with those included in the UNICEF dataset. We concluded that the extrapolated observations were within 5% of the reported observations (wherever available).

Minor issues:

1. Abstract: Conclusions “country programs will need to synergize with disease surveillance programs” – Recommend clarifying since clinical and disease surveillance are both country programs – I believe you mean MNCH clinical programs and disease surveillance programs in country.

We agree with the reviewer’s observation, and have revised the abstract text as below:

To eliminate congenital syphilis, MNCH clinical programs will need to synergize with disease surveillance programs to accurately forecast BPG demand with scale up of antenatal syphilis screening to ensure adequate treatment is available for pregnant women diagnosed with syphilis.

2. It would provide helpful context to include a baseline data table for the 11 focus countries either in main paper or additional materials – e.g. show baseline ANC coverage, HIV testing coverage, syphilis testing coverage, syphilis positivity, syphilis treatment coverage. Since country scenarios vary based on where they start with these indicators, it would put the changes over time in perspective. Additionally, in Line 253 of Discussion, you mention countries with 100% treatment among PW with syphilis but this baseline data is not shown anywhere, so would be helpful to have all data presented in discussion also available in results.

We agree with the reviewer’s suggestion on including a baseline values table. This has been included in the revised manuscript under the Methods section, under Baseline as Table 1. With the inclusion of this table before the other tables in the text, former Tables 1 and 2 are now Tables 2 and 3 respectively.

3. Table 1 and 2: Recommend changing column title from “% change” to “% difference between scenario A and B” or something similar. In my initial reading, I assumed % change referred to change over time (e.g. between 2019 and 2021) – since the two scenarios are options, and you would not change between one and the other, it may be clearer to call this a difference (in table and text).

We agree with the reviewer’s comment about there being some ambiguity in the meaning of ‘% change’ in the text and the table headings. Hence, we have appended the column heading for both tables as suggested, and added the following text to the first paragraph in the Results section:

The ‘% Change’ column indicates the difference in the demand for BPG between scenarios (a) and (b), specifically the incremental change in demand for BPG when dual tests are implemented in the screening process instead of two separate screening tests for syphilis and HIV.

4. Line 248 paragraph: Consider rewording this section on issues with syphilis testing for HIV-positive women. Describing places “where HIV prevalence precludes use of .. dual test” seems to indicate that dual test would not be appropriate for high HIV prevalence settings (like many of the countries included here), but this should not be the case. Instead, I think you assume syphilis screening would still occur among all women, including HIV-positive women who would need a separate syphilis test.

We agree with the reviewer’s comment. The following text has been added to the Discussion section in the revised manuscript:

However, country programs are still expected to test and treat all PW for syphilis, regardless of their HIV status. The WHO 2030 EMTCT targets for maternal syphilis screening and treatment are agnostic of the type of test used, and BPG demand is not affected by the same. While HIV positive PW will not be tested for HIV (which is proposed to be replaced with dual testing), country programs are expected to screen HIV pregnant women for syphilis.

---

## [Decision Letter · Decision Letter 1]

6 Aug 2021

Estimation of benzathine penicillin G demand for congenital syphilis elimination with adoption of dual HIV/syphilis rapid diagnostic tests in eleven high burden countries

PONE-D-21-12082R1

Dear Dr. Shah,

We’re pleased to inform you that your manuscript has been judged scientifically suitable for publication and will be formally accepted for publication once it meets all outstanding technical requirements.

Kind regards,

Julie AE Nelson, PhD

Academic Editor

PLOS ONE

Additional Editor Comments (optional):

Reviewers' comments:

Reviewer's Responses to Questions

**Comments to the Author**

1. If the authors have adequately addressed your comments raised in a previous round of review and you feel that this manuscript is now acceptable for publication, you may indicate that here to bypass the “Comments to the Author” section, enter your conflict of interest statement in the “Confidential to Editor” section, and submit your "Accept" recommendation.

Reviewer #2: All comments have been addressed

2. Is the manuscript technically sound, and do the data support the conclusions?

Reviewer #2: (No Response)

3. Has the statistical analysis been performed appropriately and rigorously? 

Reviewer #2: (No Response)

4. Have the authors made all data underlying the findings in their manuscript fully available?

Reviewer #2: (No Response)

5. Is the manuscript presented in an intelligible fashion and written in standard English?

Reviewer #2: (No Response)

6. Review Comments to the Author

Reviewer #2: (No Response)

7. PLOS authors have the option to publish the peer review history of their article (what does this mean?). If published, this will include your full peer review and any attached files.

Reviewer #2: No

---

## [Editor Report · Acceptance letter]

10 Aug 2021

PONE-D-21-12082R1 

Estimation of benzathine penicillin G demand for congenital syphilis elimination with adoption of dual HIV/syphilis rapid diagnostic tests in eleven high burden countries 

Dear Dr. Shah:

I'm pleased to inform you that your manuscript has been deemed suitable for publication in PLOS ONE. Congratulations! Your manuscript is now with our production department. 

Kind regards, 

on behalf of

Dr. Julie AE Nelson 

Academic Editor

PLOS ONE